# CGFusionFormer: Exploring Compact Spatial Representation for Robust 3D Human Pose Estimation with Low Computation Complexity

**DOI:** 10.3390/s25196052

**Published:** 2025-10-01

**Authors:** Tao Lu, Hongtao Wang, Degui Xiao

**Affiliations:** College of Computer Science and Electronic Engineering, Hunan University, Changsha 410082, China; ltao@hnu.edu.cn (T.L.); s221000761@hnu.edu.cn (H.W.)

**Keywords:** 2D-to-3D lifting, human pose estimation, compact spatial representation, transformer

## Abstract

Transformer-based 2D-to-3D lifting methods have demonstrated outstanding performance in 3D human pose estimation from 2D pose sequences. However, they still encounter challenges with the relatively poor quality of 2D joints and substantial computational costs. In this paper, we propose a CGFusionFormer to address these problems. We propose a compact spatial representation (CSR) to robustly generate local spatial multihypothesis features from part of the 2D pose sequence. Specifically, CSR models spatial constraints based on body parts and incorporates 2D Gaussian filters and nonparametric reduction to improve spatial features against low-quality 2D poses and reduce the computational cost of subsequent temporal encoding. We design a residual-based Hybrid Adaptive Fusion module that combines multihypothesis features with global frequency domain features to accurately estimate the 3D human pose with minimal computational cost. We realize CGFusionFormer with a PoseFormer-like transformer backbone. Extensive experiments on the challenging Human3.6M and MPI-INF-3DHP benchmarks show that our method outperforms prior transformer-based variants in short receptive fields and achieves a superior accuracy–efficiency trade-off. On Human3.6M (sequence length 27, 3 input frames), it achieves 47.6 mm Mean Per Joint Position Error (MPJPE) at only 71.3 MFLOPs, representing about a 40 percent reduction in computation compared with PoseFormerV2 while attaining better accuracy. On MPI-INF-3DHP (81-frame sequences), it reaches 97.9 Percentage of Correct Keypoints (PCK), 78.5 Area Under the Curve (AUC), and 27.2 mm MPJPE, matching the best PCK and achieving the lowest MPJPE among the compared methods under the same setting.

## 1. Introduction

3D human pose estimation (HPE) is an active research topic in computer vision that restores key point (such as joints, limbs, etc.) positions of the human body in three-dimensional space from camera images (monocular or multi-view) or from video sequences and then reconstructs the human pose. It is one of the fundamental techniques used in human behavior analysis and serves as the foundation for many fields of sensor perception application, such as motion analysis, medical rehabilitation, human–computer interaction, and virtual reality. Many of the latest methods choose 2D HPE as an intermediate to estimate 3D pose based on 2D poses (or 3D pose sequence based on 2D pose sequence), called 2D-to-3D lifting methods, which generally involves the following steps: 2D pose exploration, 2D-to-3D lifting, and feature fusion. Due to the fact that the above algorithms only take 2D poses as input, it is highly dependent on the estimation accuracy of 2D poses. Once 2D HPE fails, it will seriously affect subsequent 2D-to-3D lifting. Therefore, exploring the 2D HPE produced by offline 2D pose estimators is still a fundamental prerequisite to achieve robust 3D HPE.

The 2D-to-3D lifting methods [1,2,3,4,5] are gaining prominence in 3D human pose estimation over data-heavy approaches [6,7,8,9,10] of directly extracting 3D poses from videos. However, the effectiveness of lifting-based methods is constrained by the quality of 2D pose detection. In addition, handling a large number of temporal features also incurs significant computational costs. In practice, 2D detections are frequently degraded by occlusion, truncation, fast motion, and extreme viewpoints, producing temporally jittery joint trajectories and biased limb lengths that propagate to the lifting stage and exacerbate depth ambiguity. From an efficiency perspective, the token complexity of spatio-temporal attention scales with sequence length and joint/channel width, while frequency–domain pipelines often require dimensional expansion for alignment, increasing memory traffic and multiply–accumulate operations. Consequently, naively enlarging temporal context or stacking multiple hypotheses tends to trade robustness for computational cost, motivating compact, noise-aware spatial encodings before temporal modeling. Recently, the requirement to model long-range temporal dependencies in 2D human pose sequences makes transformers an excellent fit for 2D-to-3D lifting methods [11,12,13,14,15,16]. PoseFormer [17] introduces a widely adopted architecture comprising spatial transformer encoders and temporal transformer encoders. To counteract unreliable 2D joint detection, MHFormer [18] enhances robustness by repeatedly encoding to generate diverse pose hypotheses. This approach directly applies temporal self-attention to spatial representations, incurring substantial computational costs. PoseFormerV2 [19] adopts a computationally efficient frequency domain approach based on Discrete Cosine Transform (DCT) coefficients and low-pass filters to model skeletal sequences. However, to ensure feature alignment, it expands the dimensionality of frequency–domain features before temporal encoding, which leads to an increased computational cost.

In this paper, we propose a transformer-based CGFusionFormer to robustly estimate the three-dimensional human pose from a 2D pose sequence in a low computational complexity way, as shown in Figure 1. Our key insight is how to make full use of local and global correlations in the 2D pose sequence and to effectively extract both features, even if the 2D pose is not very accurate, and then efficiently fuse and utilize. First, we focus on exploring spatial correlation among joints and propose a compact spatial representation (CSR), which generates local spatial multihypothesis features (called CSR domain features) from part of the 2D pose sequence. Using a Body-Part Spatial Transformer Encoder (BPS) and Filter-based Multihypothesis Compression (FMC), CSR improves the robustness of spatial feature representation against low-quality 2D pose detection and reduces the computational costs of subsequent temporal encoding. Then, we design a residual network structure, hybrid adaptive fusion (HAF), which effectively integrates CSR domain features and frequency domain features to accurately estimate the 3D human pose. The frequency domain features are global frequency features of the entire 2D pose sequence obtained in the same way as in PoseFormerV2 [19]. The experimental results on the Human3.6M [20] and MPI-INF-3DHP [21] datasets demonstrate that our work achieves an outstanding trade-off between accuracy and speed with promising 3D HPE. Our main contributions are as follows:We propose a compact spatial representation (CSR) that reduces subsequent computational costs while enhancing the capability against unreliable 2D pose detection. CSR is the first attempt to explore 2D pose sequences locally while reducing subsequent 2D-to-3D lifting computational cost at the same time.We design an effective hybrid adaptive fusion (HAF) to integrate features in both the CSR domain and the frequency domain, thereby improving the accuracy of 3D pose estimation at the expense of low computational overhead. HAF enriches the residual network structure and can adapt to the general fusion of multiple features.Our CGFusionFormer achieves comparable results on two challenging datasets for 3D HPE, demonstrating a superior speed–accuracy trade-off compared to previous methods.

## 2. Related Work

Our work emphasizes robustness and explores a compact spatial representation (CSR) to achieve model lightweighting. This section delves into transformer-based techniques in 2D–3D lifting model design and analyzes and summarizes strategies for enhancing visual processing robustness, specifically in 3D human pose estimation.

### 2.1. Transformer-Based Methods for 3D Human Pose Estimation

PoseFormer is the inaugural fully transformer-based network in pose estimation, with its spatial–temporal architecture becoming a widely adopted standard in subsequent research. CrossFormer [22] refines the internal organization of spatial and temporal transformers by introducing cross-frame interaction modules and cross-joint feature interactions, thereby strengthening spatio-temporal coupling. StriderFormer [23] redesigns the temporal branch with a strided transformer encoder to mine inter-frame dependencies efficiently, while P-STMO [12] integrates multi-frame fusion into a unified spatio-temporal transformer. Beyond architecture, P-STMO [12] also adopts an MAE-style pre-training strategy [24] with random masking in space and time to let the encoder learn the joint correlation between frames and within frames.

A parallel line of work, inspired by the seq2seq paradigm [25], performs a direct sequence-to-sequence lifting from 2D to 3D. MixSTE [26] employs a recurrent spatio-temporal encoding scheme built upon PoseFormer [17], enhancing temporal attention at the joint-trajectory level so that joint-specific temporal relationships can be captured. MixSTE [26] also shows that spatial and temporal interactions are fused rather than isolated. HSTFormer [27] divides 17 joints into 6 body parts, facilitating the modeling of temporal correlation between parts. MotionBert [28] and other multitask designs further boost accuracy by leveraging temporal signals, albeit at increased computational cost.

Taken together, most transformer-based approaches chiefly advance temporal modeling—via cross-frame coupling, sequence-to-sequence lifting, body-part grouping, and pretraining—whereas compact and informative spatial encodings remain comparatively underexplored. Addressing this gap requires lightweight spatial priors that can be cleanly integrated into existing pipelines without incurring substantial computational overhead.

### 2.2. Strategies to Enhance Robustness in Computer Vision

#### 2.2.1. Multi-Hypothesis Methods

Wehrbein [29] introduces a method inspired by Normalizing Flows [30] that leverages the concept of bijectivity to map precise 3D human poses to the 2D domain. By combining heatmaps from 2D pose detectors with latent vectors preserved during the 2D–3D process, the proposed approach proves remarkably effective in handling occlusion scenarios. MHFormer [18] merges multihypothesis and transformer concepts, using stacked spatial transformers to generate varied pose hypotheses, which are then enriched through self-encoding and cross-encoding, achieving notable results with just three pose hypotheses. This method confirms that the interaction and fusion of multiple representations can significantly enhance the robustness of the model.

#### 2.2.2. Noise-Based Processing Methods

MULLER [31] applies Gaussian filtering to suppress noise while preserving key features. PoseFormerV2 [19] models skeleton sequences with low-pass filtering, removing high-frequency noise from 2D pose sequences and extracting frequency-domain cues, which markedly improves robustness. Inspired by DDPM [32], Shan et al. [33] propose a diffusion-based pipeline: Gaussian noise is progressively injected into ground-truth 3D poses, and the model iteratively denoises together with 2D inputs to restore accurate 3D poses, which highlights the value of Gaussian noise processing in robust estimation. Simo-Serra et al. [34] propagate noise from the image plane to the shape space, further corroborating the benefit of noise modeling for robustness.

Overall, noise-aware processing complements architectural advances and offers a practical route to robust 3D lifting. A promising direction is to combine compact spatial representations with simple multi-hypothesis consolidation and lightweight spatial–frequency cues, thereby reducing the burden on temporal encoders and improving resilience to inaccuracies in 2D detections under constrained computing.

## 3. Method

### 3.1. Overview

As shown in Figure 1, CGFusionFormer takes a sequence of 2D poses as input and outputs a high-quality 3D pose estimation with low computational cost in a way of combining features of local and global receptive fields.

Formally, let $P_{2d}=\left\{p_{i}|i=-K,\ldots ,-1,0,1,\ldots ,K\right\}$ represent the set of 2D poses in a sequence, where $p_{i}$ is the pose in frame *i*, 0 represents the current frame, and the objective of CGFusionFormer is to predict the 3D pose of the current frame 0:(1)$$p_{3d}=Dec\left(Haf\left(Enc_{csr}\left(P_{2d},f\right),Enc_{freq}(P_{2d},F)\right)\right),$$
where *f* is the number of frames (3, as in Figure 1) around the current frame 0 and serves as the local receptive field of $Enc_{csr}\left(\cdot \right)$, and *F* is the entire sequence length (9, as in Figure 1, F=2K+1) and serves as the global receptive field of $Enc_{freq}\left(\cdot \right)$. $Enc_{csr}\left(\cdot \right)$ and $Enc_{freq}\left(\cdot \right)$ represent modules to generate local spatial multi-hypothesis features and global frequency features, respectively. Haf(·) is the hybrid feature fusion and Dec(·) represents the regression operation to output the estimated 3D pose $p_{3d}$.

The CGFusionFormer focuses on spatial representation and hybrid feature fusion. The proposed Compact Spatial Representation (CSR) module first utilizes the Body-Part Spatial Transformer Encoder (BPS) module to obtain local spatial features from part of the 2D pose sequence (*f* frames) and then feeds them into the Filter-Based Multi-Hypothesis Compression (FMC) module. In the FMC, we incorporate multiple Gaussian filters to improve robustness and perform feature compression in the spatial domain to reduce the computational cost associated with subsequent temporal encoding. Furthermore, we integrate advanced designs such as DCT coefficients and low-pass filters to transform poses in the total sequence from the temporal domain to the frequency domain globally. The Hybrid Adaptive Fusion (HAF) module, a novel residual network, fully utilizes the features of CSR, frequency, and CSR–frequency to enable CGFusionFormer to produce accurate 3D HPE.

Although CGFusionFormer is designed to predict a 3D pose from a sequence of 2D poses, it can serve as a basic architecture for 2D-to-3D lifting methods that choose 2D HPE as an intermediate to estimate 3D pose sequences based on 2D pose sequences from monocular or multi-view camera images or videos.

### 3.2. Compact Spatial Representation (CSR)

We describe the modules used in CSR and explain how CSR reduces computational cost while ensuring feature quality and improving robustness.

#### 3.2.1. Body-Part Spatial Transformer Encoder (BPS)

Human motion can be achieved through local motions of different body parts. Therefore, accurate and independent modeling of local body parts effectively establishes the internal connections of the overall pose.

The input to BPS is indicated by $P_{0}\in R^{f\times J\times C}$ (the subset of $P_{2d}$), where *f* represents the pose frames in the receptive field, *J* represents the number of joints in each frame, and *C* represents the high-dimensional representation of each joint.

Here, we add constraints based on body part segmentation, typically by grouping the 17 joints into 5 sets according to the dynamic chain structure of a human body:(2)$$\begin{array}{c} B^{trunk}=\left\{head,neck,thorax,spine,hip\right\},\\ B^{r\_leg}=\left\{r\_feet,r\_knee,r\_hip\right\},\\ B^{l\_leg}=\left\{l\_feet,l\_knee,l\_hip\right\},\\ B^{r\_arm}=\left\{r\_shoulder,r\_wrist,r\_elbow\right\},\\ B^{l\_arm}=\left\{l\_shoulder,l\_wrist,l\_elbow\right\}. \end{array}$$

We encode these five sets of joints ($B^{trunk}$, $B^{r\_leg}$, $B^{l\_leg}$, $B^{r\_arm}$, and $B^{l\_arm}$) using a learnable dictionary $B\in R^{5\times C}$ and transform them into body embedding vectors, which is a process termed Body-Part Partitioning (BPP). We apply BPP in two components based on the vanilla spatial encoder. First, before the integration of a learnable spatial positional embedding $E_{Spos}$ into the spatial encoder (Figure 2), the introduction of BPP is essential to impose constraints on human motion representations based on body parts external to the encoder. Secondly, BPP is integrated within a multi-head self-attention mechanism, named BpMHSA (Figure 2), to enhance connections among fine-grained features. The output of the spatial encoder with L1 layers is $Z^{Spatial}\in R^{f\times J\times C}$.

#### 3.2.2. Filter-Based Multi-Hypothesis Compression (FMC)

In scenarios of low-quality 2D pose detection, Gaussian filters effectively attenuate high-frequency noise but risk over-smoothing the signal and suppressing fine-grained motion cues (e.g., rapid local motions). As discussed above, our spatial representation $Z^{Spatial}$ strengthens local correlations and preserves the granularity of local motion, partly mitigating this drawback. To further reduce the model’s sensitivity to 2D detection quality and improve robustness, we insert a set of 2D Gaussian filters that operate on $Z^{Spatial}$ within the 3D HPE pipeline. The module is implemented as a multi-branch block (Figure 3): each branch $B_{l}$ (l≤3) applies a distinct Gaussian filter to $Z^{Spatial}$ and outputs an initial hypothesis feature.

For the upper two branches, we build a hierarchy of 2D Gaussian filters by iteratively applying the kernel *g* times to obtain $W^{\left(g\right)}$ (g<3), where a larger *g* yields stronger smoothing. The bottom branch employs an identity filter $W^{G}$. Following [31], we construct inter-branch difference filters by subtracting the outputs from parallel branches, to stabilize the denoising process and preserve the fidelity of the post-smoothing feature. As the dimensionality of the feature increases, the computational cost in subsequent encoders inevitably increases. Therefore, we apply a parameter-free reduction along the channel dimension *C* to compress the hypothesis of each branch to $C^{\prime }\ll C$. In this way, FMC produces multiple hypotheses with varying levels of smoothness, which are finally fused and unfolded to form $Z^{\mathrm{CSR}}\in R^{F\times (J\cdot C^{\prime })}$.

### 3.3. Hybrid Adaptive Fusion (HAF)

As shown in Figure 4, to address the loss of detailed features of human motion due to feature compression, we design a residual structure to fuse features before and after processing through the CSR-Frequency Feature Fusion module.

In the beginning, we describe how these features are obtained. To maintain consistency with the dimensions of $Z^{CSR}$, we employ DCT coefficients and low-pass filters from [19] to transform the 2D pose sequence into the frequency domain, resulting in $Z^{Freq}\in R^{F\times (J\cdot C^{\prime })}$, which represents features in the global frequency domain because the total pose frames *F* of the sequence are used.

Subsequently, the CSR-Frequency Feature Fusion module fuses $Z^{CSR}$ and $Z^{Freq}$ to produce $Z^{CSR\_Freq}\in R^{\left(f+F\right)\times \left(J\cdot C^{\prime }\right)}$, where $Z^{CSR}$ and $Z^{Freq}$ are concatenated with different learnable positional embeddings $E_{Cpos}$, $E_{Fpos}$, respectively.

In addition, to further enhance both local and global correlations, we merge features before and after the CSR-frequency Feature Fusion module, with an increase in computational cost. Firstly, features of the CSR domain $Z^{CSR}$ and those of the frequency domain $Z^{Freq}$ are concatenated along the temporal dimension to match the size of $Z^{CSR\_Freq}$, resulting in $Z^{CSR\_Freq^{\prime }}\in R^{\left(f+F\right)\times \left(J\cdot C^{\prime }\right)}$. Secondly, $Z^{CSR\_Freq}$ and $Z^{CSR\_Freq^{\prime }}$ are spatially merged to form the integrated feature $Z^{HAConcat}\in R^{\left(f+F\right)\times \left(2\cdot J\cdot C^{\prime }\right)}$. To adaptively adjust the impact of local and global correlations during the fusion process, we introduce a balancing parameter λ into $Z^{CSR\_Freq}$. Ultimately, the final representation $Z^{HAFusion}\in R^{\left(f+F\right)\times \left(J\cdot C^{\prime }\right)}$ is derived using a fully connected layer with learnable weights $W_{F}\in R^{\left(2\cdot J\cdot C^{\prime }\right)\times \left(J\cdot C^{\prime }\right)}$. The above process can be summarized as follows:(3)$$\begin{array}{c} Z^{CSR\_Freq^{\prime }}=Concat\left(Z^{CSR},Z^{Freq}\right),\\ Z^{HAConcat}=Concat\left(\lambda Z^{CSR\_Freq},Z^{CSR\_Freq^{\prime }}\right),\\ Z^{HAFusion}=Z^{HAF}W_{F}. \end{array}$$

### 3.4. Regression Head

We employ a linear regression head, composed of a 1D convolution layer and linear mapping, to aggregate temporal information and derive the final 3D pose $y\in R^{1\times (J\cdot 3)}$ for the middle frames of the sequence, where J denotes the number of joints. Specifically, the 1D convolution layer effectively captures the temporal features within the sequence, while the linear mapping transforms these features into the final 3D joint positions.

## 4. Experiment

### 4.1. Datasets and Evaluation Metrics

*Human3.6M* [20] is a widely used benchmark for 3D human pose estimation. It was recorded in a controlled indoor studio with synchronized and calibrated multi-view RGB cameras (typically four views). The dataset contains over 3.6 million video frames of 11 subjects performing 15 everyday actions (e.g., walking, phoning) and provides accurate 3D joint annotations captured by a motion-capture system. Following the standard subject split, we use S1, S5, S6, S7, and S8 for training and S9 and S11 for testing. We adopt the Human3.6M-17 joint convention with pelvis as the root and report errors in millimeters in the camera coordinate system. The results are reported with two common metrics: MPJPE (Mean Per-Joint Position Error; Protocol #1) and P-MPJPE (Protocol #2), which computes MPJPE after a rigid Procrustes alignment [35].*MPI-INF-3DHP* [21] is also captured by camera. Compared to Human3.6M, it covers more diverse motions and scenes, including indoor green screen setups with varied lighting and a portion of outdoor environments, resulting in larger viewpoint changes, more occlusions, and more diverse backgrounds. The 3D labels are obtained via multi-view reconstruction and motion-capture pipelines with manual verification, making the dataset suitable for evaluating generalization. Following [4], we report MPJPE, PCK (Percentage of Correct Keypoints), and AUC (Area Under the Curve).

### 4.2. Implementation Details and Analysis

#### 4.2.1. Hyperparameter Settings

In our design, there are three critical hyperparameters: ksize, which denotes the kernel size of Gaussian filters $W^{1}$ and $W^{2}$; std, the standard deviation of Gaussian filters; $C^{\prime }$, the feature dimension *C* after compression in the FMC. We implement a Gaussian filter with ksize=5 and std=1 and set both $L_{1}$ and $L_{2}$ to 4 layers (Figure 1). We set *C* and $C^{\prime }$ to 32 and 24, respectively, to adequately preserve spatial features. We also employ three critical hyperparameters: the number of input frames (*f*) for the spatial encoder, the input sequence length (*F*) as the global (or expanded) receptive field, and the number of retained DCT coefficients (*n*) for preserved long-range temporal information. We set n=f when *f* is 3 or 9. When f=1, we set n to 3.

#### 4.2.2. Experimental Settings

Our experiments are carried out on a Quadro RTX 6000 GPU, with the model implemented in PyTorch [36] version 1.12.0, under the Ubuntu 22.04 operating system. During training, we set the initial learning rate at 7 × 10^−5^, incorporating an exponential learning rate decay schedule with a decay factor of 0.99. We employ AdamW [37] as the optimizer, training for 120 epochs, each with a weight decay of 0.1. To ensure fairness, following [17], we implement horizontal flip augmentation. We use CPN [38] 2D pose detection for Human3.6M following [17] and ground truth 2D detection for MPI-INF-3DHP following [39].

### 4.3. Comparison with State-of-the-Art Methods

We compare our method with state-of-the-art transformer-based approaches on two datasets, Human3.6M and MPI-INF-3DHP, demonstrating that our method achieves superior accuracy while maintaining lower computational costs.

#### 4.3.1. Results on Human3.6M

We compare our approach with representative transformer-based baselines on Human3.6M and report the results in Table 1. We vary the total sequence length *F* and the number of input frames fed into the model (denoted by *f*). We define the expansion ratio r=F/f (i.e., the ratio of the total sequence length to the number of input frames).

Our method is highly effective when the value of *f* is small and the expansion ratio *r* is large. For example, at *r* = 81 (*F* = 81, *f* = 1>), CGFusionFormer achieves an MPJPE of 47.3 mm using only 46.7 MFLOPs; by comparison, MHFormer [18] requires 342.9 MFLOPs to reach 47.8 mm MPJPE, and PoseFormerV2 [19] needs 77.2 MFLOPs for 47.6 mm MPJPE.

In addition, our proposed method achieves a superior speed–accuracy trade-off. As shown in Table 1, under similar computational budgets (approximately 75 MFLOPs) and the same sequence length (81), our method achieves an MPJPE of 47.1 mm, which is better than that of 47.6 mm obtained by PoseFormerV2 [19].

#### 4.3.2. Results on MPI-INF-3DHP

To evaluate the generality of the model, we conduct comparative experiments on the much more complex MPI-INF-3DHP dataset. The results are listed in Table 2. Our model uses an 81-frame sequence as input with n=f=9 settings, and our implementation follows [12]. The parameter settings and experimental results of the compared methods are from their original papers. Our proposed method achieves the best performance on several metrics. The results further demonstrate that our method can fully explore the features in short receptive fields and efficiently combine local and global receptive field features, making it adaptable for much more challenging 3D HPE from the 2D poses sequence.

#### 4.3.3. Robustness and Qualitative Comparisons

To evaluate the robustness of our method, we follow the experimental setup of [12] by injecting zero-mean Gaussian noise into the ground-truth 2D detections on the Human3.6M dataset. As illustrated in Figure 5, we report the performance degradation (in MPJPE) under increasing noise levels. For fair comparison, all models are evaluated using 27-frame input sequences with f=n=3. The results show that our approach not only exhibits stronger robustness under mild noise but also achieves a greater accuracy improvement than other methods even with slight noise (e.g., (sigma=3)).

Complementing these quantitative findings, Figure 6 presents qualitative comparisons to assess the robustness of 3D pose predictions under noisy 2D detections. We inject Gaussian noise into the 2D joints (with the same protocol as in Figure 5) to mimic detector drift. For fairness, all methods take 81-frame input sequences. We observe that CGFusionFormer produces more accurate and physically plausible poses, as emphasized by the annotated arrows, especially for peripheral joints (hands and feet) and the head. Compared with PoseFormer [17] and PoseFormerV2 [19], these improvements are consistent with the gains attributed to our CSR and upport the robustness indicated in Figure 5.

#### 4.3.4. Ablation Study

Table 3 shows the results of our method under different components and structures on the Human3.6M dataset. It is evident that the BPS module effectively improves accuracy with almost no increase in MFLOPs. The addition of the FMC module significantly reduces MFLOPs with only a 0.9 mm MPJPE loss compared to the baseline (1.7 × slower). Subsequently, the HAF module achieves 47.6 mm MPJPE at the cost of only four MFLOPs. The results indicate the following: (i) The substantial reduction in MFLOPs after incorporating the FMC reflects the effectiveness of decreasing computational costs. (ii) the BPS and HAF, requiring minimal computational costs, play a crucial role in enhancing performance. Notably, the baseline model is PoseFormerV2 [19], with all added modules built upon it. The inclusion of the FMC module resulted in a slight performance drop because it compressed and lost some information, as reflected in the comparison of MPJPE scores.

#### 4.3.5. Computational Complexity Analysis

To enable real-time applications of lifting-based methods, we evaluate computational complexity (Table 4). In real-time detection, because future frames are unavailable, we fill future inputs with the current frame, which leads to some performance degradation. We compare against models with nearly identical MPJPE; despite using only 46.7 MFLOPs, CGFusionFormer exhibits the smallest accuracy drop (3.7 mm MPJPE). Our method demonstrates superior stability and reliability under frame-loss scenarios compared with other robustness-oriented approaches. We also run real-time detection on an NVIDIA GeForce RTX 3050 and an NVIDIA Jetson AGX Xavier using HRNet [41] for 2D pose detection and feed the resulting sequences into subsequent 3D pose estimators. We then compare models in terms of frames per second (FPS) and MFLOPs. Our method delivers strong performance with a minimal computational budget.

## 5. Conclusions

This paper addresses core challenges such as the relatively low quality of 2D joints and significant computational costs in transformer-based 2D-to-3D lifting 3D human pose estimation methods. We propose a CGFusionFormer to estimate the 3D human pose robustly with low computational complexity from a sequence of 2D poses. To enhance the robustness of the model against unreliable 2D pose detection and reduce the computational costs associated with subsequent temporal encoding, the compact spatial representation module is proposed, which concentrates on exploring local spatial correlation in 2D pose sequences with a Body-Part Spatial Transformer Encoder and multichannel Gaussian filters-based multihypothesis compression. To further improve the robustness of the model, the Hybrid Adaptive Fusion module is designed to fully fuse features from CSR and those from the global correlation explored in 2D poses sequence. Quantitatively, CGFusionFormer attains 47.6 mm MPJPE on Human3.6M with only 71.3 MFLOPs when using 3 input frames over 27-frame sequences and with longer context reaches 47.1 mm MPJPE at similar computing. On MPI-INF-3DHP, it achieves 97.9 PCK, 78.5 AUC, and 27.2 mm MPJPE, matching the best PCK and delivering the lowest MPJPE among compared methods under the same setting. Qualitatively, under synthetic Gaussian noise, our predictions remain more physically plausible—especially for peripheral joints—than compared baselines. These results substantiate the claimed accuracy–efficiency trade-off and the robustness benefits brought by CSR and HAF. CGFusionFormer is currently designed to predict a 3D pose from a sequence of 2D poses; it can be easily extended to estimate a 3D pose sequence based on 2D pose sequences from monocular or multiview camera images or videos. Enabling real-time usage of CGFusionFormer is also part of our future work.

## Figures and Tables

**Figure 1 sensors-25-06052-f001:**
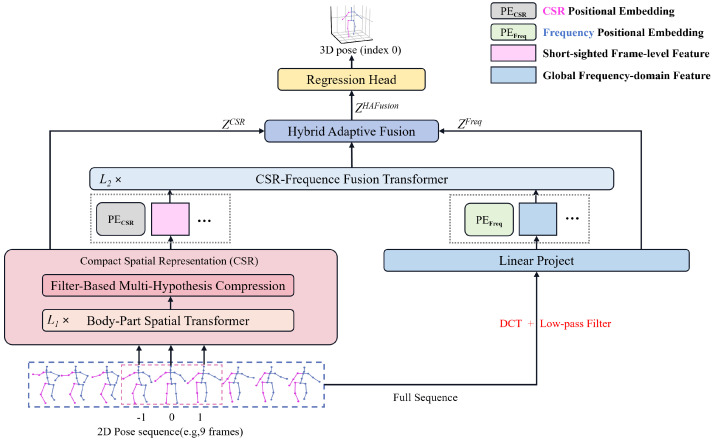
Overview of CGFusionFormer. To exemplify, given a sequence of 2D poses as input, CGFusionFormer uses 3 frames (index −1, 0, 1) to extract local CSR domain features and an entire 9 frames to capture global frequency domain features and outputs the estimated 3D human pose of frame 0 after effective and efficient hybrid feature fusion.

**Figure 2 sensors-25-06052-f002:**
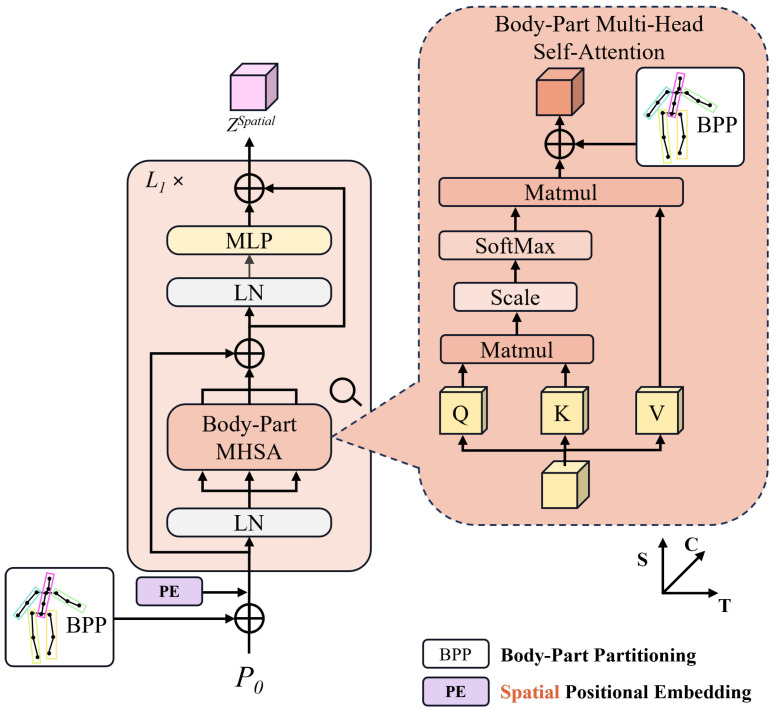
Body-Part Spatial Transformer and Body-Part Multi-head Attention.

**Figure 3 sensors-25-06052-f003:**
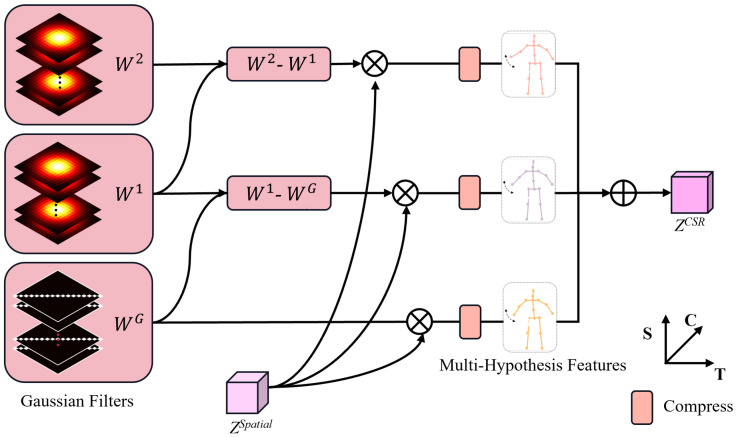
Filter-Based Multi-Hypothesis Compression.

**Figure 4 sensors-25-06052-f004:**
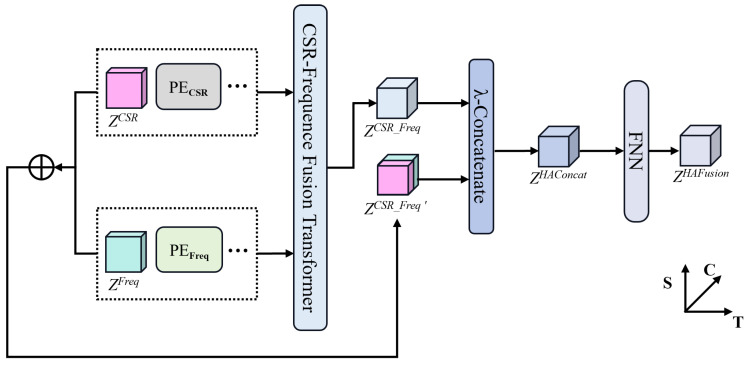
Hybrid Adaptive Fusion.

**Figure 5 sensors-25-06052-f005:**
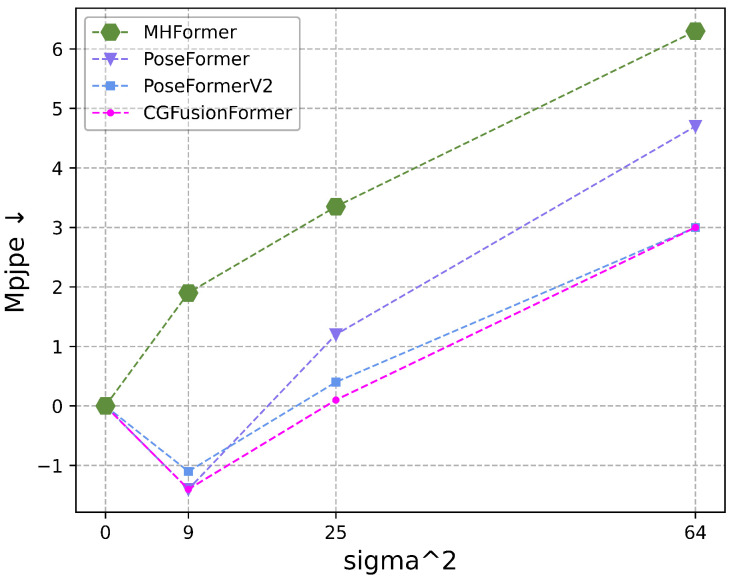
Comparison of robustness between CGFusionFormer and other methods on Human3.6M. We add zero-mean Gaussian noise with varying standard deviations to ground truth 2D detection, comparing the performance drop (MPJPE, in mm) as sigma increases.

**Figure 6 sensors-25-06052-f006:**
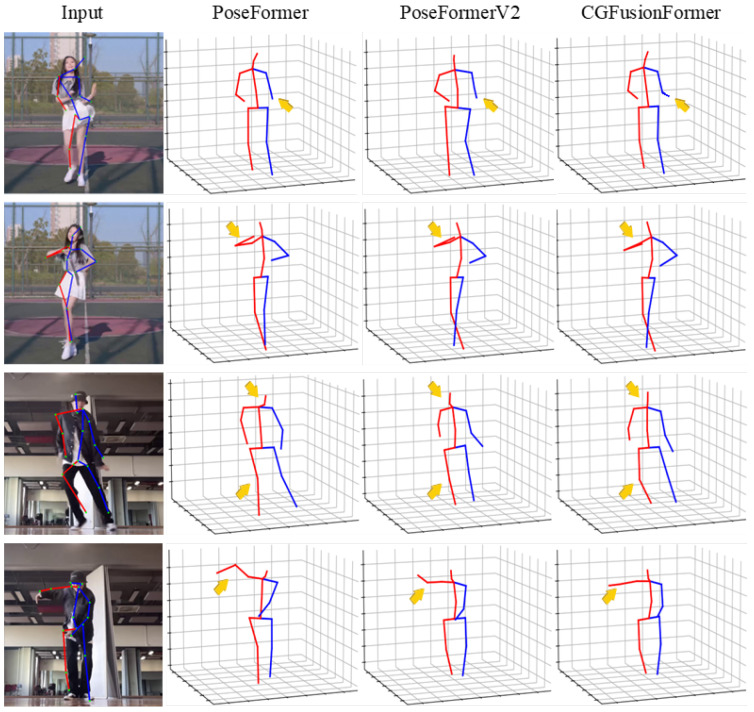
Qualitative comparison of the proposed method with PoseFormer and PoseFormerV2. We introduce Gaussian noise to the 2D detection and compare the model’s performance under conditions of 2D joint drift. Orange arrows emphasize the more accurate and realistic joint estimations in 3D poses by CGFusionFormer compared to other methods.

**Table 1 sensors-25-06052-t001:** Quantitative comparison on Human3.6M (in mm). *f* and Seq.Len. denote the local and global (or expanded) receptive field, respectively, as stated before. (*) indicates the use of an additional pre-training stage. Bolded scores represent the best outcomes.

Method	f	Seq. Len.	MFLOPs	MPJPE↓/P-MPJPE↓
PoseFormer [17]	27	27	542.1	47.0/-
StridedTrans [23]	81	81	342.5	47.5/-
MHFormer [18]	9	9	342.9	47.8/-
MHFormer [18]	27	27	1031.8	45.9/-
P-STMO [12] (*)	81	81	493	45.6/-
STCFormer [14]	27	27	2173	44.1/-
PoseFormerV2 [19]	1	27	77.2	48.7/**37.8**
CGFusionFormer (**ours**)	1	27	**46.7**	**48.5/37.8**
PoseFormerV2 [19]	1	81	77.2	47.6/**37.3**
CGFusionFormer (**ours**)	1	81	**46.7**	**47.3**/37.4
PoseFormerV2 [19]	3	27	117.3	47.9/37.4
CGFusionFormer (**ours**)	3	27	**71.3**	**47.6/37.3**
PoseFormerV2 [19]	3	81	117.3	**47.1/37.3**
CGFusionFormer (**ours**)	3	81	**71.3**	**47.1**/37.4
PoseFormerV2 [19]	9	81	351.7	**46.0/36.1**
CGFusionFormer (**ours**)	9	81	**215**	46.5/36.5

**Table 2 sensors-25-06052-t002:** Quantitative comparison with the state-of-the-art methods on MPI-INF-3DHP. (*) indicates using an additional pretraining stage. Higher PCK, higher AUC, and lower MPJPE indicate better performance. Bolded scores represent the best outcomes.

Method	Seq. Len.	PCK↑	AUC↑	MPJPE↓
Pavllo et al. [25]	81	86.0	51.9	84.0
Pavllo et al. [25]	243	85.5	51.5	84.8
Lin et al. [39]	25	83.6	51.4	79.8
Chen et al. [40]	81	87.9	54.0	78.8
PoseFormer [17]	9	95.4	63.2	57.7
MHFormer [18]	9	93.8	63.3	58.0
MixSTE [26]	27	94.4	66.5	54.9
P-STMO [12] (*)	81	**97.9**	75.8	32.2
PoseFormerV2 [19]	81	**97.9**	**78.8**	27.8
CGFusionFormer (**ours**)	81	**97.9**	78.5	**27.2**

**Table 3 sensors-25-06052-t003:** Ablation study on the baseline involving module replacement or addition. BPS represents the Body-Part Spatial Transformer Encoder, FMC denotes the Filter-Based Multi-Hypothesis Compression, and HAF stands for the Hybrid Adaptive Fusion.

		BPS	FMC	HAF	MFLOPs	MPJPE↓
Baseline	#1				117.3	47.9
+BPS	#2	✓			117.3	47.5
+FMC	#3	✓	✓		67.7	48.4
+HAF	#4	✓	✓	✓	71.7	47.6

The symbol ‘#’ denotes the experiment configuration numbers. ‘✓’ indicates that the corresponding module is included in the model variant.

**Table 4 sensors-25-06052-t004:** Comparison of computational complexity (MFLOPs), accuracy (MPJPE), and inference speed. f denotes the number of input frames per inference; Seq. Len. is the global receptive field. FPS (l) is measured on a GeForce RTX 3050 device and FPS (e) on a Jetson AGX Xavier device, and Perform. Drop (mm) refers to the performance decrease (MPJPE) caused by using the current frame in place of future frames in the input sequence. Bolded scores represent the best outcomes.

Method	f	Seq. Len.	MFLOPs	FPS (l)	FPS (e)	Perform. Drop (mm)
MHFormer [18] CVPR’22	81	81	3132.2	82.57	14.59	5.25
PoseFormerV2 [19] CVPR’23	3	81	117.3	89.93	17.74	4.3
CGFusionFormer	1	81	**46.7**	**97.95**	**18.21**	**3.7**

## Data Availability

Data are contained within the article.

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
