# Peer review of "CGFusionFormer: Exploring Compact Spatial Representation for Robust 3D Human Pose Estimation with Low Computation Complexity"

_sensors, 2025, doi:10.3390/s25196052_

Round 1

Reviewer 1 Report

Comments and Suggestions for Authors

This manuscript is well written and organized, the references are balanced, however the authors must address some minor suggestions and observations shared by this reviewer. The need to adopt and academic style to present the results in the main sections (abstract, introduction, discussion and conclusion )

Abstract section
**The authors describe ambiguous challenges related to this expression. A better description of these problems and challenges must be corrected. 

** Relevant quantitative and qualitative results can enhance the performance of abstract sections.

** Please justify for this sentence ‘’To evaluate the robustness of our method, we follow the experimental setup of [12] injecting zero-mean Gaussian noise into the ground-truth 2D detections on the Human3.6M dataset..’’, the reference by itself does not justify the use this source of noise. If possible, describe other sources to be used.

**The related works section provides a relatively comprehensive overview of existing systems; however, please clarify  the strengths and specific advantages of this system.

***The authors had relevant results, however they were not mentioned in the conclusion section. The conclusion section is not focused on quantitative and qualitative results. 

Reviewer 2 Report

Comments and Suggestions for Authors

This study proposes a Compact Spatial Representation (CSR) to robustly generate local spatial multihypothesis features from part of a 2D pose sequence. Specifically, CSR models spatial constraints based on body parts and incorporates 2D Gaussian filters and nonparametric reduction to improve spatial features against low-quality 2D poses and reduce the computational cost of subsequent temporal encoding. This study also implements CGFusionFormer with a PoseFormer-like transformer backbone. Experiments show that the proposed method produces more accurate pose estimation compared to other methods such as poseFormer or MHFormer. There are several important points for improvement:
1. The abstract summary does not yet show the experimental results of the study.
2. Provide examples of noise (both low- and high-frequency noise) that interfere with images and how the results are denoised (in image form). Does this noise significantly affect detection results?
3. Present sample: Human3.6M dataset.
4. In your proposed method, are there any parameters that can be optimized to achieve optimal accuracy? Is it possible that other methods used for comparison have also been optimized to achieve optimal results?
5. What about the baseline skeleton? Does the dataset provide this baseline as ground truth?
6. What is the calculation of the accuracy of the skeleton estimation results between the proposed method and ground truth? For example, in semantic segmentation testing, we can use IoU or dice.
7. The proposed method is claimed to have low computational complexity. What are the test scenarios?

Round 2

Reviewer 2 Report

Comments and Suggestions for Authors

The author has carefully addressed all the points raised in the review. The manuscript has been significantly improved and is now acceptable for publication.